

# Impact of forest disturbance on microarthropod communities depends on underlying ecological gradients and species traits

Davide Nardi[1,2,3], Diego Fontaneto[4,5], Matteo Girardi[6], Isaac Chini[2], Daniela Bertoldi[7], Roberto Larcher[7] and Cristiano Vernesi[2,5]

[1] DAFNAE, University of Padua, Legnaro, Italy
[2] Forest Ecology Unit/Research and Innovation Centre, Fondazione Edmund Mach, San Michele all'Adige, Italy
[3] Institute for Sustainable Plant Protection, National Research Council, Sesto Fiorentino, Italy
[4] Water Research Institute, National Research Council, Verbania Pallanza, Italy
[5] National Biodiversity Future Center, Palermo, Italy
[6] Conservation Genomics Unit/Research and Innovation Centre, Fondazione Edmund Mach, San Michele all'Adige, Italy
[7] Technology and Transfer Centre, Fondazione Edmund Mach, San Michele all'Adige, Italy

Corresponding author
Davide Nardi, davide.nardi@unipd.it

## ABSTRACT

Windstorms and salvage logging lead to huge soil disturbance in alpine spruce forests, potentially affecting soil-living arthropods. However, the impacts of forest loss and possible interactions with underlying ecological gradients on soil microarthropod communities remain little known, especially across different environmental conditions. Here we used DNA metabarcoding approach to study wind-induced disturbances on forest communities of springtails and soil mites. In particular, we aimed to test the effect of forest soil disturbance on the abundance, richness, species composition, and functional guilds of microarthropods. We sampled 29 pairs of windfall-forest sites across gradients of elevation, precipitation, aspect and slope, 2 years after a massive windstorm, named Vaia, which hit North-Eastern Italy in October 2018. Our results showed that wind-induced disturbances led to detrimental impacts on soil-living communities. Abundance of microarthropods decreased in windfalls, but with interacting effects with precipitation gradients. Operative Taxonomic Units (OTU) richness strongly decreased in post-disturbance sites, particularly affecting plant-feeder trophic guilds. Furthermore, species composition analyses revealed that communities occurring in post-disturbance sites were different to those in undisturbed forests (*i.e.*, stands without wind damage). However, variables at different spatial scales played different roles depending on the considered taxon. Our study contributes to shed light on the impacts on important, but often neglected arthropod communities after windstorm in spruce forests. Effects of forest disturbance are often mediated by underlying large scale ecological gradients, such as precipitation and topography. Massive impacts of stronger and more frequent windstorms are expected to hit forests in the future; given the response we recorded, mediated by environmental features, forest managers need to take site-specific conservation measures.

## INTRODUCTION

Forest natural disturbances are often extreme events affecting the structure of the forest, such as landslides, fires, insect outbreaks, or windstorms (*Schelhaas, Nabuurs & Schuck, 2003*). Windstorms have periodically occurred in Europe, shaping the structure of forests (*Ulanova, 2000*). However, due to climate change, windstorms are expected to increase in frequency and magnitude, causing tree uprooting at large spatial scale and damaging timber production (*Seidl et al., 2014*). Wind and other natural forest disturbances, such as fire, heavy snowfall, and pests, are well-known to increase landscape heterogeneity and to promote habitat succession and climate adaptation (*Attiwill, 1994*; *Dietz et al., 2020*). Furthermore, such disturbances usually benefit forest biodiversity by increasing niche availability and diversity at the landscape scale (*Bouget & Duelli, 2004*). However, after windstorm, fallen trees are usually logged, according to current regulations in many countries, to prevent subsequent tree mortality, mainly due to bark beetle infestations (*Leverkus et al., 2021*). Unfortunately, salvage logging causes additional disturbance on forest ecosystem leading to negative effects on soil communities because of machinery activities, removal of dead wood, and changes in soil exposure (*Thorn et al., 2017*, *2018*; *Rousseau et al., 2019*). For these reasons, predicting outcomes on the effect of windstorm in forest is challenging because site-specific and taxon-specific characteristics can influence the biological response of communities. Because of the large spatial extent, the impacts of extreme events might depend on interactions between forest disturbance and underlying ecological gradients, such as topography, forest types, and climatic gradients, thus showing a non-linear response of forest communities (*Foster, Knight & Franklin, 1998*; *Abedi et al., 2022*; *Nardi, Giannone & Marini, 2022*).

Commonly studied groups, such as pollinators, ground-dwelling arthropods, saproxylic beetles, and vertebrates, respond positively to canopy openness after wind disturbance (*Bouget & Duelli, 2004*; *Bouget, 2005*; *Thorn et al., 2016*). However, responses of soil-living communities on such forest disturbances are unclear, and more scientific effort is needed to increase our knowledge of the soil system (*Decaëns, 2010*). Previous studies showed that soil disturbances might affect a wide range of soil-living arthropods in temperate forest (*Coleman & Rieske, 2006*; *Blasi et al., 2013*; *Hartshorn, 2021*). Springtails and soil mites are amongst the most important groups of arthropods within soil ecosystems, playing a pivotal role in providing ecosystem services such as organic matter decomposition, nutrient recycling, and food webs (*Seastedt, 1984*; *Hopkin, 1997*; *Behan-Pelletier, 1999*). Biomass and diversity of microarthropod communities are important factors that can affect decomposition processes and microbial fauna (*Marshall, 2000*), as well as predator communities (*Welch et al., 2014*). Moreover, soil-living communities are known to strongly respond after changes in habitat characteristics, such as nitrogen addition and elevation (*Hågvar & Klanderud, 2009*; *Mitchell et al., 2016*; *Bokhorst et al., 2018*), canopy openness (*Perry et al., 2018*), and soil warming (*Hågvar & Klanderud, 2009*; *Thakur et al., 2023*). Declining patterns of invertebrate communities after forest disturbance are known

to be mainly associated with a decrease of resources and habitat suitability. For instance, soil perturbations, such as salvage logging, negatively affect wood-inhabiting fungi and mosses (*Thorn et al., 2018*), thus contributing to a decrease in available resources for microarthropods (*Rousseau et al., 2019*). Despite previous studies have investigated the effects of wind and salvage logging on soil microarthropod communities (*KokoŘová & Starý, 2017*; *Čuchta, Miklisová & Kováč, 2019*), the potential interactions of underlying environmental variables with windstorm disturbance and the effects on their diversity and functional guilds are still unclear.

Here, we aimed to study the effect of windstorms on forest soil using soil-inhabiting mites (mainly Mesostigmata and Oribatida) and springtails (Collembola) as model groups. We also used community DNA metabarcoding to partially overcome the difficult morphological species identification of forest microarthropods. We attempt to understand the ecological outcomes of the forest habitat changing after windstorm and the subsequent salvage logging on microarthropod communities. Due to microclimate alteration, mechanic disturbance, and resource loss, we hypothesized an overall negative effect on microarthropod communities, however, different taxa-specific responses might be expected. Here, we are interested in (I) highlighting interaction effects between underlying ecological gradients (*e.g.*, precipitation, elevation, slope) and forest disturbance, (II) describing how feeding guilds might be differently affected, and (III) assessing the role of disturbance in shaping microarthropod communities. To answer these questions, we investigated the response of springtails and soil mites using abundance, richness, trophic guilds, and species composition.

## MATERIALS AND METHODS

### Sampling design and site selection

Sampling was carried out within the Eastern Italian Alps, between Trento and Vicenza provinces. Our study area was severely hit by Vaia windstorm in October 2018 causing large windfall areas (*Chirici et al., 2019*). Windfalls mainly occurred on spruce forests and with a patchy distribution. Within our study area, windfalls were firstly mapped with high-resolution satellite images. A Digital Elevation Model (DEM, 25 m resolution) was retrieved from https://land.copernicus.eu/ and was used to compute topography-related variables, *i.e.*, elevation, slope (inclination degree), aspect (radiant distance from South). Data from 61 local climate stations were used to obtain precipitation time-series of the previous 10 years (retrieved from https://www.meteotrentino.it and https://www.arpa.veneto.it/). Mean annual precipitation values were used to compute a continuous map using kriging interpolation methods in SAGA version 7.8.2 (*Conrad et al., 2015*). These data were used in the sampling design and as explanatory variables in the models. Field inspections were carried out in candidate sites to assess operator accessibility, salvage logging operations and the presence of undisturbed spruce forest in their proximity with similar topographic and stand conditions. Only sites with a clear predominance of Norway spruce (*Picea abies* (L.) H. Karst.) forest and already logged windfalls (*i.e.*, only stumps) were considered eligible. In total, we selected 29 sites, from four geographic zones, according to independent gradients of elevation (from 1,100 to 1,950 m a.s.l.), precipitation
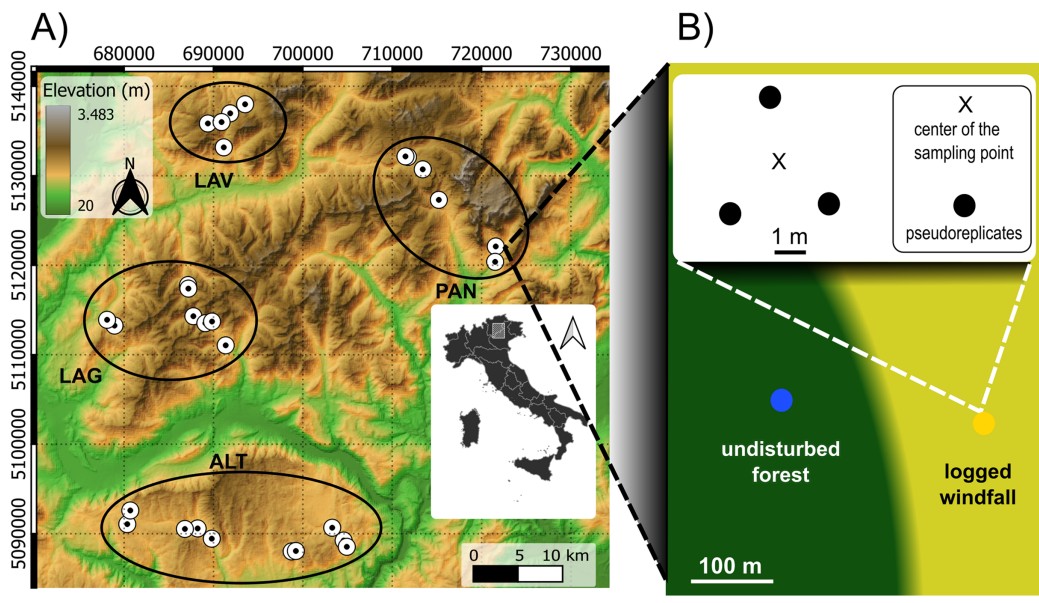

**Figure 1 Sampling design.** Sampling design consisted in 29 pair sites across different ecological gradients (Fig. 1A). The four geographic zones in the left image are highlighted with black ellipses: Lavazé (LAV), Paneveggio-Pale di San Martino (PAN), Western Lagorai (LAG), and Altopiano di Asiago (ALT). LAV and LAG have volcanic bedrock, ALT and PAN mainly carbonate. For each site, we placed two sampling points: in windfall habitat (yellow circle) and in undisturbed forest habitat (blue circle) resulting in 58 sampling points (Fig. 1B). For each sampling points three sub-samples (*i.e.*, pseudoreplicates) were taken in the neighbouring. Source elevational data: https://www.land.copernicus.eu.

(from 1,077 to 1,677 mm), slope (from 3 to 34 degree), and aspect (Fig. 1). The maximum distance between sites is about 50 km. Sites were arranged in four different geographic areas consisting of four different mountainous massifs. Vulcanic bedrocks are present in the North-West, carbonate bedrock is the predominant substrate in the South and in the East.

Then, we applied a nested pair-based sampling design. Once in the field, for each selected site, we chose two sampling points including a point in the windfall area and a point in the undisturbed forest nearby. A minimum distance of about 30 m from boundaries and ecotones was considered to avoid edge effects. The average distance between paired sampling points (within pair) was 114 m (min: 50 m; max: 265 m). Geographic positions of each sampling point were recorded with a GPS receiver. Given the precise location of the two sampling points (within each site), point-specific elevation, slope, and aspect were computed to check if there were difference between the two sampling points. In each site, the two sampling points (undisturbed forest and windfall) did not differ for elevation, slope, and aspect.

Within each sampling point, we selected three pseudo-replicated small plots of 20 × 20 cm. Pseudoreplicates were chosen few meters from the center of the sampling point. In windfall habitat, bare soil due to uprooting and machinery trails was avoided, and samples were taken from undamaged soil. For each pseudo-replicate, surface moss vegetation and soil until 5 cm depth were collected using a shovel and soil corer

(*Hishi, Kawakami & Katayama, 2022*). Percentage of moss cover was recorded by visual inspection estimates. For each sampling point, the three pseudo-replicates were mixed and pooled in a unique sample before extraction of invertebrates. Tools used for soil collection were cleaned before changing sampling point using a bleach solution. We performed two rounds of sampling during 2020, in June and in September, which is almost 2 years after the storm. For each sampling point, a sample of sieved soil (c. 10 g) without roots and moss, was preserved from the last round for organic matter, hereafter OM, analysis. OM content was analyzed by dry combustion using a CN analyzer (Primacs SNC100, Skalar, Breda, Netherlands) on dried, sieved and ground (<2 mm) samples. In detail, organic carbon released up to 400 °C (TOC 400) and residual oxidizable carbon (ROC) released up to 600 °C were analyzed in sequence using a temperature gradient from 150 °C to 600 °C after dry combustion and in presence of oxygen. OM content was then calculated as the sum of these two forms. Detailed information on sampling design is given in Supplemental Materials, Appendix A. For those locations belonging to protected areas, we obtained a sampling permission from the authorities.

## Sample preparation and DNA sequencing

In the laboratory we extracted arthropods from about 4 L of soil for each sampling site using a Berlese apparatus maintained active for 3 weeks. Animals were collected in propylene glycol and then were stored in absolute ethanol. For each sample, using decontaminated forceps and small brush, we retained only those invertebrates that were morphologically recognized as belonging to soil mites (orders Sarcoptiformes, Trombidiformes, and Mesostigmata) or springtails (class Collembola). We selected only these groups because they show similar average sizes and our sampling design is appropriate for retrieving representative communities of them. Other groups of larger arthropods, such as Carabidae, Chilopoda or Araneae, were removed since they would have been occurred randomly within the samples. Abundances of mites and springtails were counted under a dissecting microscope. All animals from each site, collected during the first and second round of sampling, were pooled together and stored in absolute ethanol until DNA extraction. Before DNA extraction, samples were dried in a Eppendorf Vacufuge centrifuge with a Qiagen vacuum pump (KNF Laboport Vacuum pump 1.0 bar) to remove ethanol. During counting and dehydration steps, some microcentrifuge tubes with 1 mL of ethanol were left open to check for potential contamination (negative controls). DNA extraction was performed using DNeasy® PowerMax® Soil extraction kit (Qiagen, Hilden, Germany) to ensure an optimum DNA extraction even in the presence of soil particles. DNA extractions of negative controls were routinely processed to assess contaminants from the lab procedure. For metabarcoding library preparation, we amplified a 313-base pair (bp) fragment of the mitochondrial DNA Cytochrome Oxidase c subunit I (COI) gene. PCR amplification was done using the mlCOIintF forward primer 5′-GGWACWGGWTGAACWGTWTAYCCYCC-3′ and jgHCO2198 reverse primer 5′-TAIACYTCIGGRTGICCRAARAAYCA-3′ (*Leray et al., 2013*). Prior to PCR amplification, a PCR condition setting up was performed in order to optimize amplicon yield. 5 μL of DNA was added to 45 μL of PCR mix, including 23.75 μL of water, 10 μL of

FlexiBuffer, 6 μL of MgCl solution (25 mM), 2 μL of each primer (10 ρmol/μL), 1 μL of dNTPs (10 mM), and 0.25 μL of Promega GoTaq Hot Start G2 (Promega, Madison, WI, USA). PCRs were run under the following conditions: one pre-incubation step of 95 °C for 2 min, followed by 45 cycles of 95 °C for 30 s, 47 °C for 60 s and 72 °C for 60 s, with a final extension at 72 °C for 5 min. All DNA amplifications were performed in a Veriti 96-Well Fast Thermal Cycler (AB Applied Biosystems).

For each sample three PCR replicates were performed, and the resulting amplicons were pooled before the purification step. Purification was carried out using QIAquick PCR purification kit (Qiagen, Hilden, Germany). Amplifications of negative controls belonging to the same PCR run were pooled. Library indexing and amplicon sequencing (300 bp) were performed at IGA (IGA Technology Services, Jacopo Linussio 51, Udine, Italy), using NexteraXT Index Kit (FC-131-1001/FC-131-1002) and an Illumina NOVASEQ6000 (300-bp paired-end mode). Please refer to the Supplemental Materials, Appendix A for the scheme of the overall sampling and sample processing protocol.

## Bioinformatic pipeline

Demultiplexed paired-end raw reads were used to retrieve ASVs (Amplicon Sequence Variants) using a custom pipeline. First, primers were cut using cutadapt (Martin, 2011) and reads without primers were discarded. Second, we used a denoising approach to retrieve Amplicon Sequence Variants (ASVs). Reads were filtered, denoised, merged, and chimeras were removed using dada2 version 1.16 package (Callahan et al., 2016).

We performed a first taxonomic assignment of ASVs using Bayesian classifier in qiime2 (https://qiime2.org) and blastn on a custom database. Unassigned, low confident (<0.97 for Bayesian confidence or <0.97 identity for blastn) or ambiguous ASVs were mapped against BOLD database (www.boldsystems.org/) to confirm their identity (January 2022). ASVs that were successfully assigned to our target groups (i.e., class Collembola and orders Sarcoptiformes, Mesostigmata, and Trombidiformes) were retained, whereas those not belonging to these groups were discarded. A further curation step was performed on sequence alignment to remove sequences with gaps or stop codons, since the amplicon region was expected to be entirely coding. AliView software (Larsson, 2014) was used for aligning and edit the sequences separately for each group.

Approaches combining denoising and clustering generally produce reliable species-like entities, which are relatively similar to those obtained with non-molecular approaches (Antich et al., 2021). However, different taxa might show different barcoding gap size, by varying intra- and inter-specific distances. Commonly used clustering algorithms (such as vsearch and swarm) depend on a priori settings and they were not optimized for a relaxed taxonomic clustering. Thus, we used an approach from DNA taxonomy, namely ASAP (Puillandre, Brouillet & Achaz, 2021), developed to identify the most appropriate distance threshold for species-level clustering from alignments of DNA sequences. ASAP clustering was performed separately for the ASVs of each group, soil mites and springtails. In order to increase the reliability of the approach, we downloaded all available species-level COI sequences of mites and springtails from BOLD database, aligned and trimmed them to our ASV datasets, and we used the large alignments (i.e., reference sequences together with

ASVs) in ASAP (https://bioinfo.mnhn.fr/abi/public/asap/). Following ASAP, ASVs were merged into Operative Taxonomic Units (OTUs) based on the barcode gap distance for each group. This aggregation step decreases intra-specific metabarcoding entities, and increases comparability with data retrieved by traditional identification. Each OTU was identified at the best taxonomic resolution according to ASV taxonomic assignment, and eventually updated according to ASAP clustering results. The final OTU table was also reduced by removing OTUs with read numbers accounting for less than 0.1% in each sample, in order to further diminish the risk of false positives in our DNA metabarcoding pipeline. The resulting incidence OTU table (only presence/absence) was used as the base for the following ecological analyses (see available data and Supplemental Materials, Appendix B).

## Statistics

First, we tested the effect of Vaia windstorm on the overall abundance of microarthropods obtained from visual counts. We used linear mixed-effect models (LMMs) with habitat type (two levels: windfall and undisturbed forest) as a predictor, in addition to elevation, precipitation, slope, aspect, OM, and their interaction with habitat type as fixed effects, with the identity of each pair of samples nested in geographic zone (four levels) as random effect. Non-significant interaction terms were removed, and models were run again. LMMs were fitted using lme4 package version 1.27.1 (*Bates et al., 2015*) in R version 4.1.3 (*R Core Team, 2022*). The response variable was log-transformed to meet model assumptions. We used DHARMa version 0.4.5 (*Harting, 2021*) and car version 3.0 (*Fox & Weisberg, 2019*) packages for checking model assumptions and collinearity among predictors. Analysis-of-variance tables were extracted with function *Anova*() in car. Spatial autocorrelation of residuals was checked using Montecarlo simulation (999 simulations) of Moran I metric. We did not find issues affecting the models (spatial autocorrelation was not significant: soil mites $P$-value = 0.40; springtails $P$-value = 0.36).

Second, we investigated the effect of Vaia windstorm on species (OTU) richness. The models we used had the same structure and rationale of the LMMs used for abundance data, separately for each of the two taxonomic groups. Model assumptions, presence of collinearity and spatial autocorrelation were checked as well. We did not find issues affecting the models (spatial autocorrelation was not significant: soil mites $P$-value = 0.25; springtails $P$-value = 0.50).

Third, we investigated the effect of wind disturbance on functional guilds, assigning each OTU to trophic niches based on literature. Although most of the OTUs were not identified to species level, feeding preferences of species are mostly the same within each family (*Krantz & Walter, 2009*; *Potapov et al., 2016*). Soil mites were divided in predators (including omnivorous) and no predators (including primary and secondary decomposers) (*Schneider et al., 2004*; *Krantz & Walter, 2009*; *Maraun et al., 2011*; *Fischer, Meyer & Maraun, 2014*; *Maaß et al., 2015*; *Schaefer & Caruso, 2019*; *Nae et al., 2021*). We preferred using such broader categorization since the trophic guilds of many species are still unknown. Springtails were divided following *Potapov et al. (2016)* in four guilds: euedaphic microorganism consumers (hereafter euedaphic), hemiedaphic microorganism

consumer (hereafter hemiedaphic), epigeic animal and microorganism consumers (hereafter animal consumers), epigeic plant and microorganism consumers (hereafter plant consumers). To test the effect of disturbance on trophic guild we used LMMs with the number of OTUs as response variable; trophic guild, habitat, and interaction between trophic guild and habitat type as predictors; site ID nested in pair ID nested in geographic area as random factor. The response variable was log-transformed for soil mite dataset. We used DHARMa and car packages as for the other models; in addition, pairwise comparisons were extracted with emmeans package (*Lenth, 2020*).

Fourth, we investigate differences in species composition (*i.e.*, beta diversity) based on OTUs occurrence data. To test the effect of ecological predictors on species composition we used Adonis analysis with marginal effects using *adonis2()* in vegan version 2.7 (*Oksanen et al., 2020*). As a response variable, we used occurrence-based Sorensen beta dissimilarity index among communities, separately for springtails and soil mites. As predictors, we used habitat type, OM, and the geographic zone and pair ID as blocks. We used 9,999 permutations for *P* value computation.

## RESULTS

Overall, we counted 15,800 soil mites (average number in each sample = 272) and 7,270 springtails (average number in each sample = 125). Regarding metabarcoding data, after sequencing and demultiplexing, the mean number of raw reads per sample was 29,588 reads ± 7,293 SD (minimum = 10,544, maximum = 44,896). After dada2 pipeline, we retrieved a mean number of 16,770 ± 4,723 SD reads per sample, constituting in total 3,041 Amplicon Sequence Variants (ASV). However, only 1,570 ASVs belonged to our target groups, which were aggregated to 441 total OTUs (289 belonging to mites, and 152 belonging to springtails) using ASAP clustering step. 201 OTUs are singletons and 78 OTUs are doubletons. Further details of OTU assignment and completeness analyses are given in Supplemental Materials, Appendix B. Both pre-PCR negative controls (extraction and sorting blanks) and PCR negative controls did not contain amplified DNA.

Regarding abundance data from morphological counts, we found that windfalls hosted overall fewer individuals than forest sites for both soil mites and springtails (Fig. 2 and Table 1). However, disturbance effect depends also on underlying ecological gradients, such as precipitation. Indeed, we found significant interactions of habitat type with annual precipitation for mites. We observed larger effect-size of disturbance (*i.e.*, decreasing abundance in windfalls) in those sites with low mean annual precipitation.

Similarly to abundance data, OTU richness decreased significantly in post-disturbance sites for both groups, soil mites and springtails (Fig. 3 and Table 2). Although we did not find significant interactions with underlying gradients, we found that richness was still shaped by ecological gradients. Soil mite richness decreased with precipitation and aspect, whereas springtail richness decreased along elevation (Table 2).

The effects of wind disturbance on trophic guilds of soil mites and springtails showed different responses. Separately for each group, we assessed the response of OTU richness for each trophic guild in interaction with habitat type. In soil mites, we found that both predators and no-predators decreased in windfalls (habitat type, *P* < 0.001) without

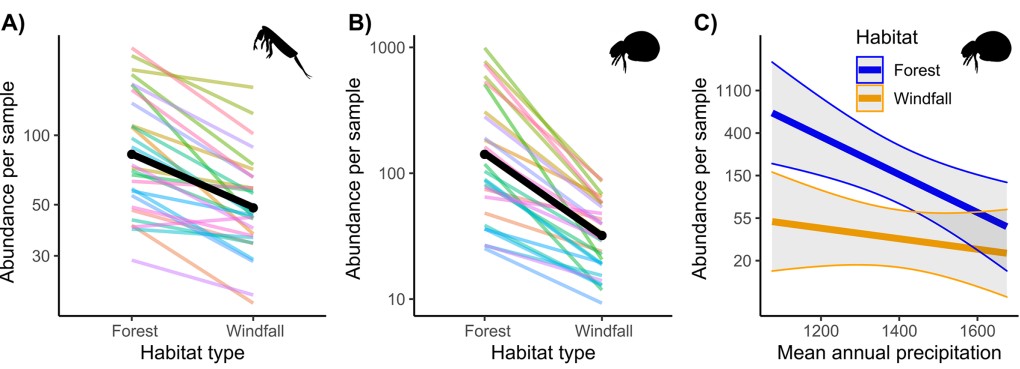

**Figure 2 Interaction effect of wind disturbance and annual precipitation on abundance data.** Results of wind disturbance (windfall habitat *vs* undisturbed forest) are shown for springtails (Fig. 2A) and mites (Fig. 2B): the overall effect is shown in black, random component (pair ID) in colour lines. Relations of soil mite abundance between undisturbed forest (blue) and windfalls (orange) with mean annual precipitation (Fig. 2C) are shown.

**Table 1 Anova-table results of abundance-based models for mites and springtails, with F and *P* values.**

**Soil mites**

|  | F | P |  |
| --- | --- | --- | --- |
| Windfall disturbance | 24.20 | <0.0001 | *** |
| Annual precipitation | 9.97 | 0.0262 | * |
| Annual precipitation: windfall | 6.21 | 0.0195 | * |
| Elevation | 1.35 | 0.2567 |  |
| Slope | 0.68 | 0.4145 |  |
| Aspect | 3.24 | 0.0800 |  |
| OM | 0.01 | 0.9280 |  |

**Springtails**

|  | F | P |  |
| --- | --- | --- | --- |
| Windfall disturbance | 5.31 | 0.0288 | * |
| Annual precipitation | 7.25 | 0.0457 | * |
| Elevation | 0.60 | 0.4490 |  |
| Slope | 0.96 | 0.3360 |  |
| Aspect | 1.80 | 0.1894 |  |
| OM | 1.41 | 0.2407 |  |

**Note:**
Number of asterisks refer to significance levels: $^*P < 0.05$, $^{***}P < 0.001$.

interactions between trophic guild and habitat type ($P = 0.66$). On the contrary, in springtails we found that soil disturbance affected epigeic plant consumer guild more than the other trophic guilds (Fig. 4 and Table 3). Similar results were obtained using moss cover as explanatory variables instead of habitat type (Supplemental Materials, Appendix C).

Regarding beta diversity patterns, soil mites showed a higher dissimilarity among samples (mean = 0.90, SD = 0.10) than springtails (mean = 0.81, SD = 0.13) (Fig. 5).

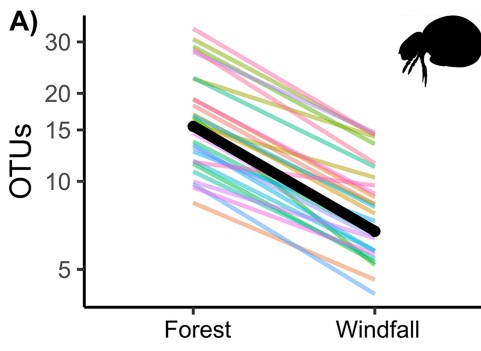

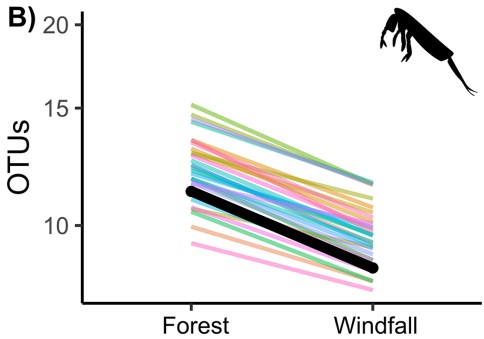

**Figure 3 Effect of wind disturbance on metabarcoding data.** OTU richness of soil mites (Fig. 3A) and springtails (Fig. 3B) compared between undisturbed forest and windfall samples. The overall effect is shown in black, random component (pair ID) in colour lines.

**Table 2 Anova-table results of OTU richness model for mites and springtails, with F and *P* values.**

**Soil mites**

|  | F | P | |
| --- | --- | --- | --- |
| Windfall disturbance | 15.28 | 0.0005 | *** |
| Annual precipitation | 7.52 | 0.0112 | * |
| Elevation | 3.21 | 0.0858 | |
| Slope | 3.10 | 0.0875 | |
| Aspect | 4.21 | 0.0478 | * |
| OM | 0.66 | 0.4215 | |

**Springtails**

|  | F | P | |
| --- | --- | --- | --- |
| Windfall disturbance | 10.27 | 0.0033 | ** |
| Annual precipitation | 1.98 | 0.1719 | |
| Elevation | 4.56 | 0.0429 | * |
| Slope | 2.65 | 0.1129 | |
| Aspect | 0.38 | 0.5433 | |
| OM | 0.23 | 0.6361 | |

**Note:**
Number of asterisks refer to significance levels: $^{*}P < 0.05$, $^{**}P < 0.01$, $^{***}P < 0.001$.

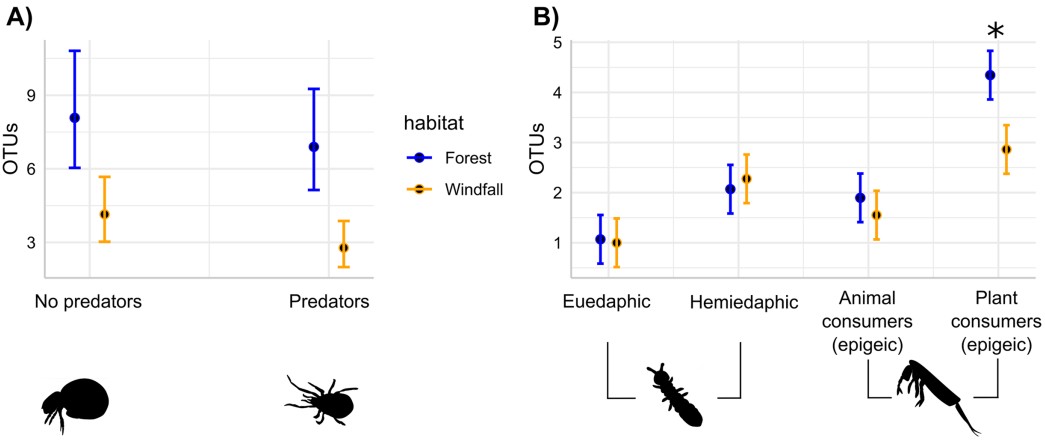

**Figure 4 Soil disturbance effect on trophic guilds.** For soil mites (Fig. 4A) and for springtails (Fig. 4B), the number of OTUs belonging to each trophic guild is compared across the two different habitats: undisturbed forest (blue) and windfall (orange). We found significant differences for both soil mites guilds (Fig. 4A), while significant differences for springtails (Fig. 4B) are shown with asterisk (*). Guilds of springtails (Fig. 4B) follow *Potapov et al. (2016)*. Complete guild names for springtails are given below following the x-axis order: euedaphic microorganism consumers, hemiedaphic microorganism consumer, epigeic animal and microorganism consumers, epigeic plant and microorganism consumers.

**Table 3 Pairwise comparisons between forest and windfall for each trophic guild of springtails.**

| Trophic guild | Habitat contrast | Estimate | SE | t | P |
|---|---|---|---|---|---|
| Euedaphic | Forest-windfall | 0.14 | 0.35 | 0.39 | 0.6940 |
| Hemiedaphic | Forest-windfall | −0.21 | 0.35 | −0.59 | 0.5552 |
| Animal consumers (epigeic) | Forest-windfall | 0.31 | 0.35 | 0.89 | 0.3765 |
| Plant consumers (epigeic) | Forest-windfall | 1.41 | 0.35 | 4.04 | 0.0001*** |

**Note:**
Estimates and standard errors (SE), t, and P values are reported. Number of asterisks refer to significance levels: ***$P < 0.001$.

We found that habitat type (*i.e.*, forest—windfall) and OM are important variables for species composition in both investigated groups (Table 4). Also, geographic zone greatly contributed to explaining species composition for springtails ($R^2 = 0.12$, P value = 0.01). However, a large part of variance remained unexplained by the models (0.77 for mites and 0.63 for springtails) suggesting that other factors might be missing. Variation partitioning using rdacca.hp package, version 1.0 (*Lai et al., 2022*) can be found in Supplemental Materials, Appendix D.

## DISCUSSION

After a massive windstorm disturbance, we investigated communities of microarthropods in undisturbed forests and windfalls using DNA metabarcoding. We assessed springtails and soil mites over a large spatial scale to test interactions between wind disturbance and underlying ecological gradients. Besides the overall decreasing abundance and species richness in post-disturbance communities, we found that windstorm effect is mediated by

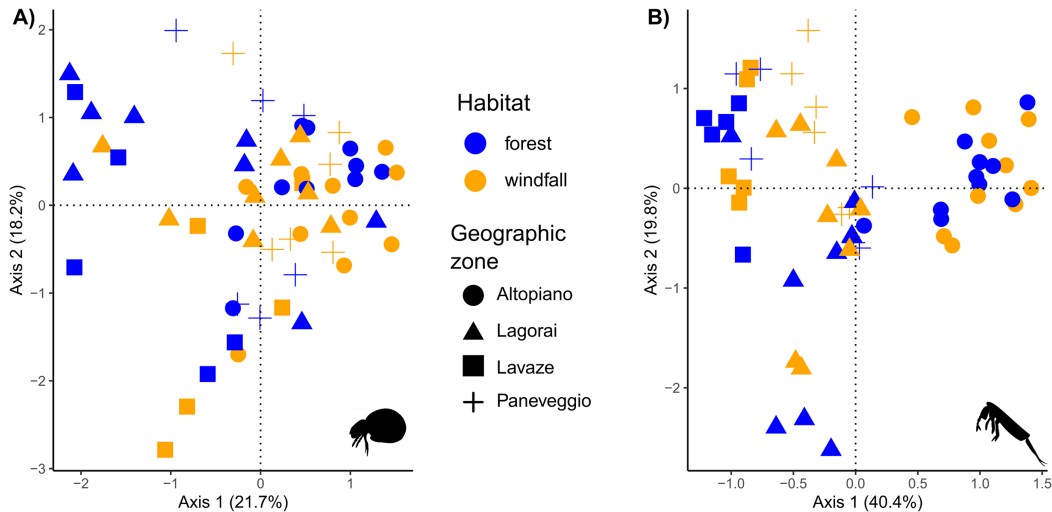

**Figure 5 Ordination analysis on presence-absence metabarcoding data.** RDA-based plots on presence-absence dissimilarity index for soil mites (Fig. 5A) and springtails (Fig. 5B). Habitat type is shown by colour (blue, forest; orange, windfall), geographic zones are shown by shapes (dot, Altopiano di Asiago; triangle, Lagorai; square, Lavazé; plus, Paneveggio-Pale di San Martino).

**Table 4 Results of Adonis model for mites and springtails.**

**Soil mites**

|  | DF | $R^2$ | F | P |  |
|---|---|---|---|---|---|
| Wind disturbance | 1 | 0.0264 | 1.62 | 0.0170 | * |
| Geographic zone | 3 | 0.0581 | 1.19 | 0.4751 |  |
| Elevation | 1 | 0.0271 | 1.66 | 0.6514 |  |
| Annual precipitation | 1 | 0.0259 | 1.59 | 0.3897 |  |
| Aspect | 1 | 0.0192 | 1.17 | 0.2845 |  |
| Slope | 1 | 0.0187 | 1.14 | 0.6955 |  |
| OM | 1 | 0.0216 | 1.32 | 0.0284 | * |

**Springtails**

|  | DF | $R^2$ | F | P |  |
|---|---|---|---|---|---|
| Wind disturbance | 1 | 0.0339 | 2.59 | 0.0001 | *** |
| Geographic zone | 3 | 0.1209 | 3.08 | 0.0107 | * |
| Elevation | 1 | 0.0297 | 2.27 | 0.0734 |  |
| Annual precipitation | 1 | 0.0291 | 2.22 | 0.2756 |  |
| Aspect | 1 | 0.0164 | 1.25 | 0.0315 | * |
| Slope | 1 | 0.0210 | 1.60 | 0.558 |  |
| OM | 1 | 0.0200 | 1.53 | 0.0360 | * |

Note:
Degrees of freedom (DF), $R^2$, F, and P values are reported. Number of asterisks refer to significance levels: *$P < 0.05$, ***$P < 0.001$.

ecological gradients and taxa-related characteristics. Topography and climatic conditions can mediate the effect of windstorm-induced disturbance, with different taxonomic groups showing contrasting responses.

## Effects of wind-induced disturbance and interactions with underlying ecological gradients

Our results showed an overall and strong decline of microarthropod abundance and richness in windfalls, compared to near undisturbed forest, suggesting that soil perturbations could strongly modify microarthropod communities and, possibly, their ecological functioning (Figs. 2 and 3). Our results are consistent with previous studies based on morphologically identified microarthropod communities showing negative effects of windstorm and salvage logging on springtails (*Čuchta, Miklisová & Kováč, 2019*; *Sławski & Sławska, 2019*), soil mites (*Lóšková et al., 2013*; *Wehner et al., 2021b*), and proturans (*Sterzyńska et al., 2020*). Literature evidence showed that post-disturbance communities are mainly constituted by few high-adaptable species, such as *Tectocepheus* sp. (mites) and some species belonging to Entomobrydae (springtails) (*Maraun et al., 2003*), or by species with disturbance-adapted strategies, such as parthenogenetic reproduction (*Wehner et al., 2021a*). On the contrary, other studies found that richness of springtails increased in disturbed plot (*Urbanovičová, Kováč & Miklisová, 2010*; *Urbanovičová, Miklisová & Kováč, 2014*), and others found divergent responses in abundance and species richness (*Čuchta, Miklisová & Kováč, 2012*). On the one hand, these differences might depend on the time span of the studies since the response may change depending on time from disturbance (*Čuchta, Miklisová & Kováč, 2019*). For instance, after intense drought mites were slower than springtails in recovery processes (*Lindberg & Bengtsson, 2005*). On the other hand, different responses might also depend on interactions with site-specific conditions. Indeed, here we showed how underlying ecological gradients (*i.e.*, precipitation) beyond the main effect of disturbance, can change the response of the communities, especially in complex environments such as alpine mountain forests.

Indeed, we showed that interaction between canopy openness (*i.e.*, wind-induced disturbance) and precipitation influenced the response of soil communities to disturbance. We found higher effect-size of disturbance on mite abundance at low mean annual precipitation (Fig. 2). In other words, drier conditions (*i.e.*, low annual precipitations) showed higher loss of mite individuals after forest canopy removal. Precipitation gradient can strongly influence soil biota, especially for forest ecosystems where aridity might have detrimental effects on soil-living communities (*Blankinship, Niklaus & Hungate, 2011*). Indeed, moisture is known to be an important factor for microarthropods (*Hopkin, 1997*; *Lindberg & Bengtsson, 2005*), and habitat alterations resulting in drier conditions might exacerbate negative effects on microarthropod communities. Despite analyzing soil moisture might be more informative for soil arthropods than large-scale rainfall, moisture is expected to change with forest disturbance and cannot be separated from the disturbance gradient. On the contrary, two close sampling points are expected to have the same average rainfall, and rainfall effect can be tested independently. We suggest that disturbance-driven soil impacts might exacerbate under climate change in the future, especially for forest ecosystems. Despite soil disturbance after windstorms could be intensified by topography conditions, such as steep slopes, due to soil instability (*Mitchell,*

*2013*), we did not find interactions with other local factors. Probably our large-scale approach cannot effectively catch all the effects at the local scale.

## Simple and interacting effects of wind-induced disturbance on trophic guilds

Microarthropods could exhibit a large differentiation in trophic niche and feeding preferences, even within taxonomic groups, such as oribatid mites (*Schneider et al., 2004*) and springtails (*Potapov et al., 2016*). However, after disturbance, ecosystems could exhibit trophic web declining with collapsing of certain trophic niches. For instance, oribatid communities in plantations showed missing niches in food webs compared to rainforest habitat (*Krause et al., 2021*), and functional guilds of microarthropods are differently affected by soil warming (*Thakur et al., 2023*). Here, we assessed the effect of soil disturbance on trophic guilds to test whether forest disturbances can influence functional traits in the disturbed soils. We found that functional guilds of soil mites and springtails are differently affected by windstorm. In particular, we observed that the richness of soil mites strongly declined regardless of trophic guilds. Previous studies found that the proportion of detritivore oribatid mites might depend on forest management (*Farská, Prejzková & Rusek, 2014*). Many reasons could be advocate for explaining the lack of this effect in our data. First, trophic guild assignment was based on taxonomic identifications since we could not directly measure it. Since taxa might be assigned with different taxonomic resolution, those with low taxonomic resolution cannot be specifically addressed to precise trophic niche, but we used general trophic niche (mostly at family level). Second, we used a coarse trophic partitioning, in order to include taxa with wide range of trophic preferences (generalists or poorly studied taxa). Moreover, previous studies showed that the same species can slightly change its feeding preferences under certain environmental conditions (*Melguizo-Ruiz et al., 2017*; *Maraun et al., 2020*). Specific approach, such as isotopic analyses might provide further insights on the effect of soil food webs after wind disturbances in temperate forests (*Maraun et al., 2011*).

On the contrary, we found significant interactions between habitat type and trophic guild in springtails. Epigeic plant consumers were the most affected guild in windfalls. This guild-specific response might be due to the dramatic decrease of moss cover in windfalls (Supplemental Materials, Appendix C), as moss represents the main feeding substrate. Indeed, higher insolation due to canopy removal changes micro-conditions and salvage logging might largely contribute by exacerbating the desiccation process (*Waldron et al., 2014*). Our results agreed with previous studies emphasizing that habitat specialist species of springtails, such as bryophilous species, are the most sensitive ones to environmental changes after forest clearings (*Urbanovičová, Miklisová & Kováč, 2013*). Actually, the combined effect of canopy openings and salvage logging led to huge changes in litter and superficial soil layers (*i.e.*, organic layer), vegetation, and moss cover (*Rumbaitis del Rio, 2006*), largely affecting those species related to litter microhabitats, in particular specialists. Despite our studies focused on short-term effects, similar long-term studies revealed that

springtail communities can remain affected by the disturbance even for decades (*Sławski & Sławska, 2019*). Hence, we suggest that our findings on springtail perturbations in windfall communities might have long-term outcomes. Here, we showed that mite and springtail communities might have different response to soil disturbance: an overall decrease for soil mites and guild-related losses for springtails.

## Effects of wind-induced disturbance and distance on species composition

In our analyses soil mites and springtails responded to different spatial scales. These results agreed with literature, suggesting that dispersal might play a role in post-disturbance communities with group-related responses (*Rousseau et al., 2019*). Composition of soil mite communities exhibited a high dissimilarity among samples, suggesting that small-scale conditions, such as the presence of microhabitats, are important drivers for species assemblages in forest communities (Fig. 5). Our results showed that habitat type and OM were the most important factors contributing to differences in species composition for mites. Windfall and undisturbed forest hosted completely different communities and differences in mite communities might occur at very small scale, even within the same habitat type. On the contrary, the distance at large spatial scale (*i.e.*, geographic area) did not play an important role in the model. Moreover, our model revealed a relatively high amount of unexplained variance, probably due to the high diversity and dependence of microscale conditions, which are not always available for such large-scale studies. Our findings agreed with the low mobility and strict environmental niche preferences of this group (*Lehmitz et al., 2012*). Thus, even short distances may actually reflect huge differences in terms of environmental variables, as similar studies found that soil mite communities can differ in species composition within few metres (*Dong et al., 2017*).

Much more similar communities are expected in springtails since they have higher mobility than mites, especially those species with a well-developed *furca* (*Potapov et al., 2020*). Indeed, in our model the main factor shaping differences between communities was geographic zone, while habitat type, aspect, and OM were less important (Fig. 5). Our findings suggested that geographic distance had a higher explanatory power than soil disturbance on differences in species assemblages for springtails. Our result might depend on two main reasons. First, besides the relevance of microenvironmental parameters, within continuous habitat types springtail communities might be shaped by geographic distance due to their limiting dispersal (*Arribas et al., 2021*). Second, the higher explanatory power of geographic distance than habitat type might still reflect a short-term response to disturbance. Here, we sampled only one forest type (*i.e.*, pure spruce forests) just 2 years after the disturbance, thus it is not surprising that geographic zone (*i.e.*, related to historical patterns) is still the most important factor. On the other hand, small-scale variables, such as habitat type, still play a great importance as predictors for species composition, in agreement with previous studies (*Salmon & Ponge, 2012*; *Sterzyńska & Skłodowski, 2018*; *Arribas et al., 2021*).

### Limitations

Our method allowed us to derive comparable results among samples and thus drawing general conclusions on the ecological patterns after massive wind disturbance in forest. Unfortunately, we could only assess the compound effect of windthrow and salvage logging, since no-logged areas were not accessible. Besides this study limitation, salvage logging is extensively carried out in most of the forests in Europe, being the main management strategy after severe windthrows. Moreover, this dataset should not be considered a complete overview of the forest-living communities, because metabarcoding has some limitations. First, different markers might provide slightly different communities. Second, our samples are mainly constituted by specimens, but they included also soil particles, making them between pure tissue samples (*i.e.* bulk sample) and a soil sample (*i.e.* environmental sample). Finally, the taxonomic resolution depends on the reference database, which is often affected by the lack of completeness for the studied groups.

## CONCLUSION

Our results suggested that windstorm-induced disturbance might have detrimental effects on soil microarthropods in temperate forests, at least in the short term. Negative effects on soil functionality and diversity could affect bio-mediated soil processes, such as decomposition (*Marshall, 2000*). Furthermore, changes in habitat vegetation and soil decomposer communities may have bottom-up effects, also affecting upper trophic guilds, such as predatory macroinvertebrates (*Laigle et al., 2021*). Natural disturbances are extremely complex phenomena and interactions between wind-induced disturbance and other large-scale or micro-scale underlying ecological gradients might reduce or increase the effect on arthropod soil communities. Hence, since increasing soil disturbance in forests, as well as drying climate, are expected in the near future, conservation actions and mitigation measures should be prioritized by forest management after natural disturbances. Finally, we showed how DNA metabarcoding could be used as an integrative approach, to help identify and monitor taxonomic groups that are poorly known but highly ecologically relevant in forest ecosystems.

## ACKNOWLEDGEMENTS

We thank Nancy Gálvez-Reyes, Maria Sterzynska and an anonymous reviewer for improving the first version of the manuscript with useful comments.

### Funding

The work has been carried out in the frame of a PhD grant supported by the University of Padova (DAFNAE-DOR), Fondazione Edmund Mach and the Institute for Sustainable Plant Protection—National Research Council (IPSP-CNR). There was no additional external funding received for this study. The funders had no role in study design, data collection and analysis, decision to publish, or preparation of the manuscript.

## Grant Disclosures

The following grant information was disclosed by the authors:
University of Padova (DAFNAE-DOR).
Fondazione Edmund Mach.
Institute for Sustainable Plant Protection—National Research Council (IPSP-CNR).

## Competing Interests

The authors declare that they have no competing interests.

## Author Contributions

- Davide Nardi conceived and designed the experiments, performed the experiments, analyzed the data, prepared figures and/or tables, authored or reviewed drafts of the article, and approved the final draft.
- Diego Fontaneto analyzed the data, prepared figures and/or tables, authored or reviewed drafts of the article, and approved the final draft.
- Matteo Girardi performed the experiments, authored or reviewed drafts of the article, and approved the final draft.
- Isaac Chini performed the experiments, authored or reviewed drafts of the article, and approved the final draft.
- Daniela Bertoldi performed the experiments, authored or reviewed drafts of the article, and approved the final draft.
- Roberto Larcher performed the experiments, authored or reviewed drafts of the article, and approved the final draft.
- Cristiano Vernesi conceived and designed the experiments, performed the experiments, authored or reviewed drafts of the article, and approved the final draft.

## Field Study Permissions

The following information was supplied relating to field study approvals (*i.e.*, approving body and any reference numbers):

Field experiments in protected areas were approved by Parco Paneveggio Pale di San Martino.

## Data Availability

The raw sequences, datasets, R scripts are available at Zenodo: Nardi Davide, Diego Fontaneto, Matteo Girardi, Isaac Chini, Daniela Bertoldi, Roberto Larcher, & Cristiano Vernesi. (2023). Impact of forest disturbance on microarthropod communities depends on underlying ecological gradient and species traits (1.0) [Data set]. Zenodo. https://doi.org/10.5281/zenodo.8185893.

## Supplemental Information

Supplemental information for this article can be found online at http://dx.doi.org/10.7717/peerj.15959#supplemental-information.

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
