# Peer review of "Impact of forest disturbance on microarthropod communities depends on underlying ecological gradients and species traits"

_PeerJ, doi:10.7717/peerj.15959_

## Round 0.1 · original submission · Major Revisions

Dear Authors,

After receiving feedback from three reviewers, it has been determined that your study presents an interesting and novel approach. However, two of the reviewers have raised concerns regarding the level of detail provided in certain aspects of the methodology, as well as some minor observations. Therefore, significant revisions are required before the paper can be accepted for publication.

Sincerely,

Armando Sunny.

·

Basic reporting

Nardi and coauthors present an interesting paper exploring the microarthropod communities of spruce forests with a focus on the windfall-forest mechanisms of community assembly. The authors collected invertebrates in 5 cm depth of soil and separated samples in Berlese traps and employed COX1 metabarcoding to generate Collembola and mites community data. As well as a review of abundance and richness, the analyses mostly focused on abundance and richness compared with GLM analysis with elevation, precipitacion, functional guilds and spatial distance.

The turnover component of beta diversity, examining the extent to which different Collembola and mites groups form distinct similarity clusters in space (using NMDS and Adonis). The authors concluding that the abundance and richness decreased in windfall, and also disturbance, precipitation, and plant-feeder trophic guilds drive the abundance and richness of communities.

The relative novelty of this work is the ability to exploit the comprehensive data generated through metabarcoding to obtain ASV information for an entire community of Collembola and mites, then leverage this information to examine abundance and richness community processes in detail. Furthermore, the study sites are certainly understudied for these sorts of questions.

Experimental design

However, this dataset is somewhat limited, which does narrow the scope of this paper in some regards, and the paper largely lacks any critical evaluation of the methods employed and the potential limitations these impose on the ability to draw wide conclusions about community assembly mechanisms. In general, while the conceptual and theoretical aspects of this paper appear well-presented and well-reasoned, the methodology and results could do with some revision to both improve justification for some methodological aspects and improve clarity for the reader.

Furthermore, the authors report that they carried out extraction with soil kit and used replication of amplification and didn't quantify DNA genomic. In the absence of information, I assume that equimolar amounts of these fractions were combined; however, the authors should report this and also justify whether this appropriately ensures that small species are equally likely to be recovered through metabarcoding.

Validity of the findings

no comment

Additional comments

L26. “Impacts on soil microarthropods are poorly studied,”. Just because something has not been done, is not in itself a good rationale for doing the study. Please clarify the rationale. What larger conceptual questions can be addressed by studying processes within sites?

L.35 What do “after disturbance” refer to here? Do you have a record of a “before”? Since the collection sites are very close, I would like to know how they approach this part?

L37. “intact forest” How do you know what an intact forest is? if the same hurricane happened to windstorms.

L41. What are the micro-scale factors?

L45. I suggest choosing one “springtails” or “Collembola”.

L48. I suggest starting with a sentence at the beginning, including what is a disturbance or the cause of a disturbance and then go with the example.

L69. “is a poorly” Please clarify the rationale. The suggestions above (line 26) might help here.

L70-72 “Blasi et al., 2013” You refer to several articles and only one investigation is mentioned here. Please add other papers.

L72-75. Are there any studies that indicate that springtails and mites are good indicators of disturbance?

L87. Please clarify the biological rationale. The suggestions above (line 26, 35) might help here.

L90. Define if the work is with eDNA metabarcoding or community DNA metabarcoding. What is the difference between both? and in this case, define with which method you approached the study.

L102. Please add meters above sea level.

L104-105. “causing large damages” what damage?

L107-116. Could you please explain for which analysis these data were used?

L117. What is the minimum and maximum distance between sites?

L120. “sampling design” could you describe the design better? Even in the Supplementary material there could be an outline to address this. Specify the number of samples that belong to the intact forest and the disturbed area.

L122. Please put the GPS mark.

L127 What was the distance between pseudoreplicas?

L129-130. Why only 5 cm? if there is a reference that supports that.

L133-134. How to address this variation in data collection in your analysis? Did you do any analysis that reports the difference between that variation in time? If not, could you explain how you address the statistical heterogeneity between samples due to climatic variation between the two times? For example Barsoum et al. 2019 mentions that metabarcoding was able to detect temporal changes between weeks in the composition of the communities.

Barsoum, N., Bruce, C., Forster, J., Ji, Y. Q., & Douglas, W. Y. (2019). The devil is in the detail: Metabarcoding of arthropods provides a sensitive measure of biodiversity response to forest stand composition compared with surrogate measures of biodiversity. Ecological Indicators, 101, 313-323.

L148-150. “We selected only these groups because” This section must be associated with the justification of the introduction. Although if you are justifying the size, please put a reference of why.
For example, what happens when the samples have significantly different body sizes in metabarcoding? please justify.

L153. Please add the vacuum mark.

L156. So far I have read that it is extracted with the Qiagen kit for tissue and blood. Why did you use the soil kit? explain the biological sense. Is there any work that supports the use of this microarthropod tissue soil kit? since they could lose biodiversity with the incorrect use of the kit.

L159-169. If you prepared the libraries, please add this information in the supplementary material as you did this part. For example, if you did the method with a single PCR or (a second step for the ligation of the barcodes or tag?
Did you follow any previously published protocol for library preparation? Which was?
If they used "overhang" adapters?. What barcodes did you use for tagging the libraries? (this in case there was a second PCR).

L173-174. Sequencing was PE or SE? Please provide this detail here. Also the full name of the place where they did the sequencing with NOVAseq.

L176. For reproducibility of the science, I suggest placing the bioinformatics pipeline and ecological analysis scripts in a GitHub repository.

L182. Please add the date the blastn was made.

L187. Describe the method to detect and remove stop codons, and what reference were you based on? Why is it important to stop codons?

L207. Community tables of filtered ASV were then transformed into incidence (presence/absence) data for downstream analyses. This is right? Please clarify this.

L210-220. I suggest doing LMG analysis including dissimilarity.
Above all, to know if the distance influences the structure of the community. Because the distance can cover the effects of the other characteristics.

L248-250. How many reads did they get in each direction for each sample?

L253. Is metabarcoding a good technique to recover microarthropod diversity? since they obtained 279 for mites and 152 for collembola. Does it have to do with the marker they chose or the kit they used? justify please.

L263. I suggest a completeness analysis of OTU richness, to check if the data is sufficient for further analysis.

L259. “underlying abiotic factors” which are?

L278-281. “geografic great contribute to explain species composition” How do you explain that it has a very low R? Here I suggest doing an LGM with the beta diversity (dissimilarity) with the variable data from Table 4. In this way we will know if the distance between sites and/or the characteristics shape the community structure.

L294. “at large spatial scale” Please put in methods the minimum and maximum distance between the sites in methods.

L296. “near intact forest sites” I suggest adding in the description of the sites in methods. Please standardize this term throughout the article.

L311. What “other underlying gradients”?

L373. I suggest doing an IBD (Isolation by distance) with dissimilarity. Similar analysis above in line 278.

L378. Clarify if only for Collembola.

L391-392.To ensure that the geographic distance influences the composition of the community, I suggest a GLM analysis with both geographic variables and other characteristics from Table 4 for both orders.

Reviewer 2 ·

Basic reporting

This study examined the effect of windfall on the soil microarthropod community, by comparing abundance, OTUs richness and ecological traits. The study found strong impact of disturbance and salvage logging on Acari and Collembola communities.
The morphological determination of individuals was replaced with the metabarcoding. This novelty approach enabled the processing of a large number of localities, which can clearly be considered a strong point of the project. I appreciate the fact that the authors have included the trait assessment to link the shifts in species composition to ecosystem functioning. Unfortunately, the low taxonomic resolution given by the use of metabarcoding did not allow the authors to use more specific traits that would answer more complex ecological questions.

English is clear and unambiguous, but still needs important improvement in the right use of comma, prepositions etc.
Introduction, text structure and references are ok. Context is well described.

Experimental design

Experimental design is ok, only minor comments on description in the Material and methods section

Validity of the findings

Unfortunately, there are important data missing in the paper. It is stated in lines 207-208 “The resulting OTU table was used as the base for the following ecological analyses (Supplementary B).” In the Supplementary B, there are only the pictures showing taxonomic resolution achievable for mite and springtail OTUs and no table. The table with a list of OTUs is an essential part of this type of study. It should contain list of OTUs, assigned taxa, accession numbers of the sequences, closest match in GenBank with its accession number, similarity and assigned ecological guild with the reference.

Additional comments

ABSTRACT
ok

INTRODUCTION
1) lines 60-63 Sentence is unclear and is hard to follow.

MATERIAL AND METHODS
2) line 138 sieving is < 2 mm, not < 0.02 mm
3) line 156 What was the weight of the sample? What amount of microarthropods was used for the analysis?
4) line 161 How were the primers marked by the tags? and how many tags were used?
5) line 163 How many ng of DNA was used? The volume of the solution is not sufficient information, because the solution can have a various concentration.
6) and instead of end

DISCUSSION
7) line 315 The effect of slope was not significant. (Table 1)
8) line 321 correct comma in the word gradient,s
9) lines 321-323 The sentence about ecological gradients is placed in the middle of the paragraph dealing with humidity and precipitation and has no connection with the surrounding sentences.
10) lines 357-358 The effect of moss cover on the guilds of springtails can be statistically evaluated. The authors have all the necessary data and the statistical analysis would support their findings.
11) lines 396-399 The findings that springtails are more affected with geographic zone than by the windstorm and logging are interesting and deserve a more detailed explanation. For example, which mentioned historical patterns can underlie the differences? These information would be useful for the readers not familiar with Italian geography and history. Are there some other differences among the four study areas? What about bedrock?

TABLES AND FIGURES
12) Use consistent description throughout the tables. Windfall & Habitat in Table 1, Precipitation & Annual precipitation, p & P in statistics

·

Basic reporting

Manuscript presents effect of forest disturbance considered as compound effect of windthrow and salvage logging on the soil microarthropods communities. Authors examined Acari (Mesostigmata and Oribatida) and Collembola as model group using molecular approach (DNA metabarcoding) and OUT-s units as a proxy of the traditional systems of biological classification. They tested the effect of forest disturbances on abundance, richness, species composition, and functional guilds. Research on soil microarthropod community was conducted across 4 windfall forest areas, differing in precipitation and topography, within the Eastern Italian Alps two years after a massive windstorm “Vaia”. Within study region 29 pairs of windfall-forest and intact sites across gradients of elevation, precipitation, aspect and slope were examined. As result authors documented interaction between large scale ecological gradients (topography and precipitation), micro-scale factors and forests disturbances (windstorm + salvage logging) which might reduce or increase the effect on soil microarthropods communities. In conclusion, they stated that DNA metabarcoding could be used as an integrative approach to monitor soil microarthropods taxa, such as Collembola and Acari, response to forest disturbances. They also documented that forest disturbances are extremely complex phenomena and interactions between wind-induced disturbance and other large-scale or micro-scale underlying ecological gradients might reduce or increase the effect on arthropod soil communities.

The presented paper is clear, introduction and background well indicate the context of the research;
Literature is well referenced and relevant. Figures are also relevant. The ms. structure conforms to PeerJ standards. However, some fragments require clarification. First important point is that examined effect of Vaia windstorm includes two aspects of “forest disturbances” : natural wind-induced disturbance and tree-clearing disturbance, and that windfall habitat is also related to logged sites. Therefore, for example, the description on the x-axis in Fig. 2 seems inappropriate - because the study sites included windfall and logged habitats.
Also examined effect of “forest disturbance” on trophic guilds identified in Collembola should be consistent with the Potapov at al. 2016 - which lists the following groupings and functional leagues: epigeic plant and microorganisms consumers, epigeic animal and microorganisms consumers, hemiedaphic microorganisms consumers and euedaphic microorgnsms consmumers.

Detailed remarks
Line 33 – the results showed not only wind-induced disturbances but also forestry logging
Line 93-96 – interaction effect between underlying ecological gradients and forest disturbance – it is unclear what kind of ecological gradients?
Line 115 – if besides the similar topographic conditions, the age of disturbed and intact spruce stands were similar?
Line 117-118 authors introduced name "geographic zones" for 4 study areas within the Eastern Italian Alps as meaning of independent gradients of elevation, precipitation, slope and aspect. However, a geographical zone is defined as a region divided by latitude or longitude. In Material and Methods there is also no indications what kind of differences were among studied regions/areas; Figure 1 - there lack of coordinates.
Line 224 – investigated the effect of disturbance – what kind?
Line 232 – please correct to edaphic
Figure 2 – y-axis abundance – did you present abundance per sample ?
Line 268 – effect of soil disturbance on trophic guilds - what was used as a predictor of soil disturbance?

Experimental design

no comment

Validity of the findings

no comment

Additional comments

The conducted research using operational taxonomic units to analyze the impact of forest disturbances on assemblages of soil microarthropods is pioneering in this respect.

---

## Round 0.2 · Minor Revisions

Dear authors,

While one reviewer has already accepted the manuscript, it appears that the other reviewer believes that some additional minor corrections are required.

I would like to express my gratitude to the reviewer for taking the time to provide their valuable observations. I eagerly await the corrected version of the manuscript.

Best regards,

Armando Sunny

·

Basic reporting

The authors have effectively addressed the previous concerns.

Experimental design

The methods are sufficiently detailed and provide enough information for replication purposes.

Validity of the findings

The provided underlying data is comprehensive, exhibits statistical rigor, and has been effectively controlled.

Additional comments

I have only a few minor concerns to raise.

Minor comments:
I commend the authors on their efforts towards ensuring data reproducibility. However, I suggest depositing the scripts in the Dryad repository or GitHub, and setting the raw sequences in some SRA project from NCBI to enhance data availability.

Regarding Figure 5, I appreciate the authors' utilization of NMDS, but it lacks results on the ANOSIM test. It is important to note that ANOSIM demonstrated significant dissimilarity differences between the Forest and Windwall groups.

In line 189, I recommend replacing "Library preparation" with "Library indexing..." Although the indexing of each library using the Nextera XT kit primers (PCR2) was conducted by the company, you were responsible for the initial library preparation. Therefore, I suggest making a change in line 174 to include "For metabarcoding library preparation,..." before "We amplified a 313-base pair."

·

Basic reporting

The manuscript presents the correct version of the work concerning of the effect of forest disturbance considered as compound effect of windthrow and salvage loggingc on the soil microarthropods communities. Although the authors make a number of improvements compared to the first round of the review the presented version still needs minor corrections and clarifications before being accepted for printing.

First, please indicate that richness concept is related to OUT-s richness throughout the text or clearly specify in the material and methods chapter what meaning the term richness used throughout the text.


Other detailed remarks :
Line 37 – 38 – please indicate that richness concept is related to OUT-s richness.
Line 89 – 92 – fragment unclear – does the “temperature factor” and “warming” is related to soil warming?
Line 93-94 - - the subject is missing in the sentence; declining patterns of what ?
Line 286 – 287 - please explain whether B-diversity was based on Otus units?
Line 326-327 - sentence is formulated in such a way that it does not represent a result, but assumptions that should be presented n the material and methods chapter.
Line 350 – 357 - does the comment refer to Otus richness? However, the papers cited below noting changes in the species richness of soil microarthropods were all based on classical taxonomic analysis of species composition. How does this result correspond with the results of these studies?
Line 377 – size of disturbance? How size of disturbance was estimated? Did you related size of disturbance to canopy openness ?
Line 398 - subtitle is unclear, the effect of what?
Line 429 – 432 – fragment unclear and needs improvement; bryophilous species of springtails are specialist species.
Line 432-435 – please use accepted names of soil horizon layer.
Line 445- 448 – part needs to be changed. Please explain what kind of the processes producing species assemblages can be identified on a small scale and what and what could be the reason such a large diversity of mite communities in forests.

Experimental design

no comment

Validity of the findings

no comment

Additional comments

Thank you very much for the all explanations and corrections. I think that after clarifying the above inaccuracies and minor doubts, the article will be very interesting for soil zoologists.

---

## Round 0.3 · accepted · Accept

Dear Authors,

I am delighted to share the exciting news that both reviewers have expressed satisfaction with the remarkable improvements made to the manuscript. As a result, I am thrilled to announce that your work is now ready for publication.

We extend our heartfelt gratitude for considering PeerJ as the platform to showcase your captivating and thought-provoking research.

Best Regards,

Armando Sunny

·

Basic reporting

Impact of forest disturbance on microarthropod communities depends on underlying
ecological gradient and species traits
D. Nardi, D. Fontaneto, M. Girardi, I. Chini, D. Bertoldi, R. Larcher, C. Vernesi

Many thanks to the authors for all corrections and clarifications. The revised version of the manuscript submitted for review may be accepted for publication. I found only one stylistic inconsistency line 399 - 402 (numbering in tracked version) …”Our results agreed with previous studies emphasizing that habitat specialist species of springtails, such as bryophilous species, are the most sensitive ones to environmental changes after forest clearings (Urbanovičová, Miklisová & Kováč, 2013) maybe ….that habitat specialist among springtails, such as bryophlous species…”

Experimental design

no comments

Validity of the findings

no comments

Additional comments

no comments